# The Signaling Pathway of TNF Receptors: Linking Animal Models of Renal Disease to Human CKD

**DOI:** 10.3390/ijms23063284

**Published:** 2022-03-18

**Authors:** Irina Lousa, Flávio Reis, Alice Santos-Silva, Luís Belo

**Affiliations:** 1Associate Laboratory i4HB-Institute for Health and Bioeconomy, Faculty of Pharmacy, University of Porto, 4050-313 Porto, Portugal; irina.filipa@hotmail.com (I.L.); assilva@ff.up.pt (A.S.-S.); 2UCIBIO—Applied Molecular Biosciences Unit, Laboratory of Biochemistry, Department of Biological Sciences, Faculty of Pharmacy, University of Porto, 4050-313 Porto, Portugal; 3Institute of Pharmacology & Experimental Therapeutics & Coimbra Institute for Clinical and Biomedical Research (iCBR), Faculty of Medicine, University of Coimbra, 3000-548 Coimbra, Portugal; freis@fmed.uc.pt; 4Center for Innovative Biomedicine and Biotechnology (CIBB), University of Coimbra, 3004-504 Coimbra, Portugal; 5Clinical Academic Center of Coimbra (CACC), 3000-075 Coimbra, Portugal

**Keywords:** CKD, inflammation, TNF-alpha, TNFR, biomarkers

## Abstract

Chronic kidney disease (CKD) has been recognized as a global public health problem. Despite the current advances in medicine, CKD-associated morbidity and mortality remain unacceptably high. Several studies have highlighted the contribution of inflammation and inflammatory mediators to the development and/or progression of CKD, such as tumor necrosis factor (TNF)-related biomarkers. The inflammation pathway driven by TNF-α, through TNF receptors 1 (TNFR1) and 2 (TNFR2), involves important mediators in the pathogenesis of CKD. Circulating levels of TNFRs were associated with changes in other biomarkers of kidney function and injury, and were described as predictors of disease progression, cardiovascular morbidity, and mortality in several cohorts of patients. Experimental studies describe the possible downstream signaling pathways induced upon TNFR activation and the resulting biological responses. This review will focus on the available data on TNFR1 and TNFR2, and illustrates their contributions to the pathophysiology of kidney diseases, their cellular and molecular roles, as well as their potential as CKD biomarkers. The emerging evidence shows that TNF receptors could act as biomarkers of renal damage and as mediators of the disease. Furthermore, it has been suggested that these biomarkers could significantly improve the discrimination of clinical CKD prognostic models.

## 1. Chronic Kidney Disease—A Public Health Issue

In the last decade, chronic kidney disease (CKD) has been recognized as a global public health problem, due to its increasing incidence and prevalence rates [1,2]. Additionally, CKD is a significant contributor to early morbidity and mortality worldwide, as well as an important risk factor for cardiovascular diseases (CVD). In 2017, CKD was the 12th leading cause of death, globally, rising from 17th in 1990 [3].

CKD is a pathological condition that results from a gradual and permanent loss of renal function over time, characterized by the presence of kidney dysfunction and injury markers, over a period of at least three months. According to the ‘2012 Kidney Disease: Improving Global Outcomes’ (KDIGO) guidelines, the severity of CKD is classified into five stages, according to glomerular filtration rate (GFR) and urinary albumin excretion [4]. Increased CKD severity is indicated by lower GFR and/or increased albuminuria levels.

The etiology of CKD depends on the setting, with diabetes and hypertension being the two major causes of kidney injury in developed countries [3]. However, irrespective of the primary disease cause, CKD initiation and progression involves different pathophysiological pathways leading to kidney function decline [5], which involves a complex interaction between hemodynamic, metabolic, immunologic, and inflammatory mechanisms.

CKD is associated with a decreased quality of life, increased risk of hospitalization, cardiovascular complications and mortality, independently of other risk factors [1,3,6,7]. Importantly, CKD and its related comorbidities are largely preventable and manageable, if detected at an initial stage. Thus, early identification of CKD is essential, not only to predict and prevent CKD progression, but also to further improve patients’ survival and reduce associated morbidities. Hence, more sensitive and earlier biomarkers of detection are necessary to achieve that goal, since the traditional biomarkers only increase when a significant filtration capacity has already been lost and kidney damage is advanced [8].

Several studies in the literature suggest that activation of inflammatory processes in the early stages of CKD drives kidney function impairment [5], meaning that the assessment of inflammatory markers might help in earlier diagnosis of CKD. Associations between biomarkers of inflammation and changes in GFR have been widely reported. Moreover, inflammation is a risk factor for CKD-associated morbidity and appears to contribute to cardiovascular mortality in CKD patients [9,10,11,12].

## 2. Inflammation as an Essential Component of CKD

The persistent low-grade inflammatory status that characterizes CKD plays a key role in the pathophysiology of the disease. Inflammation starts early in the onset of renal diseases [13,14] and worsens with disease progression [15], being particularly marked in hemodialysis patients [16]. Interestingly, inflammation can be identified either as a trigger or a consequence of CKD. The etiology of inflammation is multifactorial and can result from a primary cause of disease (diabetes, obesity) [17], from renal dysfunction comorbidities (uremia, metabolic acidosis, intestinal dysbiosis, vitamin D deficiency, oxidative stress) [17,18,19], and/or from dialysis procedures (intercurrent infections and thrombotic events) [20].

Inflammation is a well-established risk factor of both morbidity and mortality in CKD patients [21,22,23], leading to renal function deterioration and fibrosis. CKD patients present low to moderate levels of circulating inflammatory mediators [24,25] as a result of a deregulation of their synthesis, increased release, and/or impaired renal clearance [15,26]. It is broadly accepted that inflammation plays a role in CKD progression, but the association between disease initiation and the establishment of inflammation is debatable. Glomerular hypertrophy, endothelial dysfunction, podocytes damage, proteinuria, and tubular cells injury are some of the identified kidney insults that can trigger the development of inflammation [27].

The initial inflammatory response occurs to overcome renal injury, promote tissue remodeling and wound healing. However, when this process outreaches the physiological limit, a chronic inflammatory state may arise, with undesirable systemic consequences [14]. The dysregulated immune response results in a continuous activation of inflammatory mediators, contributing to renal scarring and fibrosis [24], the final common pathological manifestation of renal diseases.

The inflammatory state is characterized by activation of inflammatory cells, releasing an array of acute phase proteins, cytokines, and chemokines [19,25], which are able to interact with renal parenchymal cells and resident immune cells, and trigger the recruitment and activation of circulating monocytes, lymphocytes, and neutrophils, into renal tissue [13,14]. The activation of inflammatory response and the infiltration of inflammatory cells induce cellular transdifferentiation into myofibroblasts, which are responsible for the production and deposition of extracellular matrix components and cytoskeletal components, which leads to renal remodeling. In renal fibrosis, myofibroblasts seem to be derived from different cell types, such as tubular epithelial cells, interstitial fibroblasts, macrophages, as well as pericytes and endothelial cells [28]. The imbalance in matrix formation/degradation leads to accumulation of an extracellular matrix, which might lead to glomerulosclerosis and/or tubulointerstitial fibrosis and a consequent GFR decline [25,29]. Under chronic inflammatory activation, resident kidney cells exhibiting a proinflammatory phenotype, coupled with the activated immune cells, are responsible for perpetuating the ongoing inflammatory process, leading to renal fibrosis. Once renal fibrosis sets in, CKD progression is irreversible, irrespective of the initial cause [29]. Kidney hypoxia/ischemia, inflammation, and oxidative stress are simultaneously a cause and an effect of renal damage and fibrosis. Those events form a vicious cycle in CKD progression.

Cytokines and acute phase proteins are simultaneously key mediators and biomarkers of inflammation. Even though their circulating concentrations show a tendency to increase with the worsening of disease, the rate and magnitude of the increase depends on the molecule itself. It has been shown in several CKD models, that the classical proinflammatory signaling pathway, the NF-ĸB system [18,30], is activated by multiple inflammatory mediators, mainly by tumor necrosis factor alpha (TNF-α).

## 3. The TNF Signaling Pathway

Tumor necrosis factor alpha (TNF-α), also known as TNF superfamily member 2 (TNFSF2) or simply TNF, is a pleiotropic cytokine that can mediate the inflammatory response, regulate immune function by promoting immune cells activation and recruitment, and may trigger cell proliferation, differentiation, apoptosis, and necroptosis [31]. TNF-α is primarily produced by activated immune cells, and its increase in the circulation can be detected within minutes after the pro-inflammatory stimuli [32]; TNF-α can also be expressed by activated endothelial cells [33], fibroblasts [34], adipose tissue [35], cardiac myocytes [36], and neurons [37]. Abnormally elevated production, and/or sustained higher values of TNF-α, have been associated with autoimmune diseases, such as rheumatoid arthritis, multiple sclerosis, inflammatory bowel diseases [38,39], and chronic inflammatory disease states, such as sepsis, CKD, obesity, and diabetes [35,40,41].

TNF-α can be found in two bioactive homotrimeric forms: as a 26 kDa transmembrane peptide, or as a 17 kDa soluble form that is released into circulation upon cleavage by the metalloproteinase TNF-α converting enzyme (TACE) [31,42]. The pleotropic actions of TNF-α are mediated by either one of its two TNF receptors, TNFR1 and TNFR2 [32], which engage shared and distinct downstream signaling pathways; therefore, both exhibit common and divergent biological functions. While TNFR1 is basally expressed across all human cells [43] and is more efficiently triggered by soluble TNF-α, TNFR2 is mostly expressed in immune cells, endothelial cells, and neurons and has more affinity for the TNF-α membrane-bound form [44]. Besides its independent functions, TNFR2 acts as a ligand presenting TNF-α to TNFR1, potentiating its response [45]. Through the activity of TACE enzymes, TNFR1 and TNFR2 membrane receptors can also be converted into soluble forms, which act as antagonists of TNF-α [46].

TNF-α exerts both homeostatic and pro-inflammatory roles. However, TNF-α binding to TNFR1 mostly promotes inflammation and tissue injury [47], while binding to TNFR2 has been mainly implicated in immune modulation and tissue regeneration. TNFR2 is also essential for epithelial-to-mesenchymal transition and cell proliferation [47,48]. Thus, the immunoregulatory functions of TNF-α involve multiple mechanisms and depend on the regulation and relative expression of the two receptors, as well as their shedding [49].

TNFR1 and TNFR2 present different intracellular domains [50] that can interact with common and diverse downstream signaling molecules [47]. Figure 1 illustrates the TNFR1 and TNFR1 signaling pathways. The role of each receptor is context-dependent and can also be cell or tissue specific.

### 3.1. TNFR1 Signaling Pathways

The pathways triggered upon TNFR1 activation are better known. TNFR1 contains an intracellular death domain (DD) that, in the absence of ligand, interacts with a cytosolic silencer of death domains (SODD) [50]. Upon binding to TNF-α, the inhibitory protein SODD is released and the DD of TNFR1 is recognized by the TNF receptor-associated death domain, TRADD, which recruits two additional adaptor proteins, TNF receptor-associated factor 1 or 2 (TRAF1/2) and receptor interacting serine/threonine-protein kinase 1 (RIPK1) [51,52,53]. The assembling of different signaling pathways that activate distinct downstream responses will depend on the ubiquitination state of RIPK1 [54]. Thus, RIPK1 is the major regulator of the cellular decision between TNF-mediated pro-survival signaling or death.

Ubiquitinated RIPK1 allows the activation of complex I, comprising TRADD, RIPK1, TRAF2, cellular inhibitor of apoptosis protein 1 or 2 (cIAP1/2), and linear ubiquitin chain assembly complex (LUBAC) [55,56]. Both cIAPs and LUBAC promote poly-ubiquitination of RIPK1 [55,56], which leads to the recruitment of two complexes: transforming growth factor-beta-activated kinase 1 (TAK1) complex, comprising TAK-binding proteins (TAB) 2 and 3; and inhibitor of kappa B kinase (IKK) complex, involving two kinases IKKα and IKKβ, and the regulatory subunit NF-κB essential modulator (NEMO, also known as IKKγ) [57]. The recruitment of these two complexes (TAK1 and IKK) leads to the activation of mitogen activated protein kinases (MAPKs) and the canonical NF-κB pathway.

The activation of the IKK complex requires NEMO ubiquitination, by LUBAC [56,58], and IKKβ phosphorylation, by TAK1 [55,59]. In turn, the phosphorylated IKKβ initiates phosphorylation and proteosomal degradation of NF-ĸB inhibitor (IκB), unmasking the p65 subunit of NF-κB and enabling the translocation of the NF-κB heterodimer, composed of a p65 and a p50 subunit, into the nucleus, where it activates the transcription of various proinflammatory, anti-apoptotic, and pro-survival genes [60]. Another TAK1-dependent mechanism that upregulates proinflammatory gene expression involves the phosphorylation of mitogen activated protein kinases (MAPKs), such as c-jun kinase (JNK) and p38 [59,61], which further induces activator protein-1 (AP-1) transcription factor [62].

Alternatively, TNF-α binding to TNFR1 can induce two types of programmed cell death, apoptosis or necroptosis, when death-inducing signaling complexes (IIa, IIb, or IIc) are assembled in the cytosol [43]. TNF-TNFR1 mediated NF-ĸB signaling induces cell survival and requires polyubiquitination of RIPK1 bound to TRADD [63,64]. Therefore, when RIPK1 is not ubiquitinated, it dissociates from complex I, which favors the formation of death complexes. In these NF-ĸB inhibited conditions, TRADD recruits Fas-associated death domain (FADD) [53], a pro-caspase 8 dimer, and a heterodimer of pro-caspase 8 and the long form of cellular FLICE-inhibitory protein (c-FLIPʟ) [57], forming complex IIa.

The depletion of the cIAP1 and 2 also reduces, or prevents, RIPK1 ubiquitination [65], resulting in apoptosis, through complex IIb. This cytoplasmic complex, formed by nonubiquitinated RIPK1 and FADD, recruits RIPK3, pro-caspase 8 dimer, and c-FLIP-pro-caspase 8 heterodimer, and induces apoptotic cell death, similarly to complex IIa [57].

Furthermore, the aggregation of RIPK1 and RIPK3 leads to the activation of mixed lineage kinase domain-like protein (MLKL) [66,67], through complex IIc. Several mechanisms by which phosphorylated MLKL induces necrotic cell death have been proposed [68], such as mitochondrial fragmentation and/or plasma membrane rupture with a subsequent influx of positively charged ions.

Ubiquitination of the proteins involved in TNFR1-signaling cascades has a major role in determining TNF-induced downstream outcomes. Ubiquitination status of RIPK1 determines whether TNF-TNFR1 signaling mediates cell survival or apoptosis, since RIPK1 ubiquitination prevents complex IIa and IIb from assembling. Several ubiquitin-modifying proteins that act on RIPK1 have been identified. The ubiquitin-modifying enzyme A20 is able to bind and remove polyubiquitin chains from RIPK1 and LUBAC, blocking NF-ĸB activation. Cylindromatosis (CYLD) is another deubiquitylating enzyme that acts on several proteins, such as TRAF2, RIPK1, and IKKγ, to regulate the NF-ĸB and JNK pathways [43,69]. In addition, cellular degradation or depletion of cIAPs prevents RIPK1 ubiquitination.

### 3.2. TNFR2 Signaling Pathways

Unlike TNFR1, TNFR2 does not have a DD, being unable to recruit TRADD [53]. Upon TNF-α binding, TNFR2 interacts directly with TRAF 1 or TRAF2, which recruits cIAP1 and 2, along with LUBAC [70,71]. Accordingly with the events triggered by TNFR1 signaling, the ubiquitin chains formed by LUBAC allow the recruitment of TAK1 and IKK complexes; therefore, activating the canonical NF-κB signaling pathway.

However, TNFR2 may also trigger non-canonical NF-κB signaling [72], by promoting activation of the NF-κB inducing kinase (NIK) [73]. In the absence of stimuli, NIK is ubiquitinated by intracellular TRAF/cIAPs complexes, and undergoes proteasomal degradation. However, upon TNF-α binding, the subsequent recruitment of these complexes by TNFRs, leads to NIK stabilization and activation. Activated NIK phosphorylates and induces the processing of p100, a protein that acts as an IκB-like molecule, which allows the nuclear translocation of p52/RelB [73]. This evidence confirms earlier studies that showed that the TNFR2 signaling involved in NF-ĸB activation occurs independently of TNFR1 signaling, which highlights distinct molecular pathways not shared with TNFR1 [74].

Furthermore, TNFR2 is able to activate JNK and protein kinase B (Akt) pathways [75,76]. TNFR2-mediated JNK activation seems to be TRAF2-dependent [76]. TNFR2 associates with apoptosis signal-regulating kinase-1 (ASK-1), an upstream MAPK critical for JNK activation [77]. TNFR2 mediated endothelial/epithelial protein tyrosine kinase (Etk) activation, subsequently stimulates phosphatidylinositol 3-kinase (PI3K) and its effector Akt, promoting pro-survival and reparative cascades [78]. These pathways are likely to be involved in the TNF-dependent activation of mesenchymal stem cells and T cells [44].

In endothelial cells, TNFR2 also signals through interferon regulatory factor-1 (IRF1), inducing interferon-β (IFN-β), promoting the transcription of inflammatory cytokines and monocytes recruitment during a TNF-induced inflammatory response [79].

TNFR2 can induce cell death indirectly by crosstalk with TNFR1. Depletion of TRAF2 by TNFR2 inhibits the NF-kB and MAPK signaling pathways mediated by TNFR1, favoring the formation of death complexes [80].

When TNFR1 and TNFR2 are co-expressed in the same cells, intracellular crosstalk between both signaling pathways seems to be mainly shaped by intracellular constraints, such as the availability of downstream effectors of each pathway, such as TRAF2 and ASK-1 [81,82]. However, there are other factors that contribute to the complexity of this cross-talk, such as the differential expression of both receptors in different cell types and the fact that the two signaling pathways are linked by positive and negative feedback mechanisms [44].

## 4. Involvement of TNF Receptors on Renal Deterioration

The inflammation pathway driven by TNF-α is important in the pathogenesis of CKD [83,84]. However, the role of TNF-α and its receptors in renal diseases is not completely clarified. Upon an inflammatory stimulus, TNF-α was shown to be overproduced in podocytes, mesangial cells, proximal tubules, glomerular cells, and also in infiltrating macrophages [84], amplifying the overall injury response. While TNFR1 is generally found in glomerular and peritubular endothelial cells, TNFR2 expression in renal cells has been shown to be transcriptionally induced after renal injury [84,85].

In this review, we summarize the more important results from published studies on the contribution of TNF-α and its receptors to the development and/or progression of CKD (Table 1 and Table 2). A search in Pubmed was conducted, including animal and human studies, using the keywords “renal disease”, “chronic kidney disease”, or “CKD”, and the biomarkers name “TNF-α”, “TNFR1”, and “TNFR2”, to search the title and/or abstract. From the retrieved articles, and after title and abstract screening, we selected studies that evaluated the validity of these biomarkers in CKD diagnosis and prognosis, in different renal disease models and patients with different backgrounds. Furthermore, we searched for additional publications in the references of the selected articles.

### 4.1. Studies Addressing TNF-α and TNFRs in Animal Models

Animal studies are the primary source of evidence for the role of TNF-α in the development of kidney diseases (Table 1). In the classical 5/6 nephrectomy CKD model, NF-ĸB is activated and other proinflammatory genes are upregulated [86]. The systemic administration of TNF-α in rat models of anti-glomerular basement membrane antibody-mediated nephritis worsened the severity of glomerular injury by increasing neutrophil influx, albuminuria, and the prevalence of glomerular capillary thrombi [83]. TNF-α blockade reduced proteinuria, inflammation status, and renal scaring in mice [87] and rat [88] models of glomerulonephritis. It was also shown that TNF-α blockade prevented the development of crescents in a rat model of crescentic glomerulonephritis [88] and reduced renal tubular cell apoptosis, caspase activity, and several markers of renal fibrosis, in a model of unilateral ureteral obstruction [89,90].

Studies addressing the deletion of TNFR1 and/or TNFR2 genes, in animal models, also illustrated the contributions of the TNFRs in the pathophysiology of kidney diseases. The deletion of TNFR1 was associated with an increase in GFR, in an angiotensin II-induced model of hypertension [91]. Data from the same study showed that renal TNFR2 mRNA expression is increased in hypertensive TNFR1 knockout mice, along with increased urinary albumin excretion, compared to wild type mice and to TNFR1 knockout mice without induced hypertension. The authors suggested that TNFR2 has a leading role in the development of albuminuria [91]. Accordingly, TNFR2 knockout mice subjected to immune complex-mediated glomerulonephritis did not exhibit increased albuminuria and were protected from renal injury, despite preserving intact the immune system response [92]. In a model of unilateral ureteral obstruction, both TNFR1 and TNFR2 knockout mice showed a significantly reduced relative volume of the cortical interstitium, in the obstructed kidney, compared with the wild-type mice, as a result of the decreased deposition of pro-fibrotic proteins [93]. Additionally, the individual knockout of TNFR1 or TNFR2 resulted in decreased inflammation, demonstrated by the reduced activation of the NF-κB pathway. TNFR deletion was found to have comparable favorable effects in kidney disease development in several other animal studies [94,95,96,97].

TNFR participation in diabetic kidney disease has been the subject of specific research. Previous studies reported that the TNF-α inhibition protects against tubular injuries [97] and prevents renal hypertrophy [98] in diabetic rats. A diabetic mice model treated with a TNF-α inhibitor, Etanercept, showed improvements in albuminuria, decreased expression of inflammatory molecules, and decreased macrophage infiltration into the kidney [99]; renal levels of TNFR2, but not TNF-α or TNFR1, were decreased compared to non-treated mice [99]. The authors suggested that diabetic nephropathy is predominantly associated with the inflammatory action of TNF-α via the TNFR2 pathway. Other works also demonstrated that the administration of TNF antagonists inhibits salt retention, renal hypertrophy [98], and albuminuria [100], suggesting that TNF inhibition may slow the progression of diabetic nephropathy.

Transcriptomics further showed that both oxidative stress and inflammation play a role in the pathogenesis of CKD, and are correlated with cellular alterations that lead to systemic complications [101]. In ischemia–reperfusion mice models, proximal tubule cells at a late injury stage that mimic chronic progression confirmed a marked activation of the TNF, NF-κB, and AP-1 signaling pathways [102].

**Table 1 ijms-23-03284-t001:** Association of TNF-α and TNF receptors with renal dysfunction and disease in animal models.

Year	Study Model	Methods	Study Outcomes	Reference
1989	Anti GBM nephritis rat model	Pretreatment with human TNF-α	Pretreatment of rats with TNF-α increased the glomerular neutrophil influx and exacerbated glomerular injury, judged by the increased albuminuria and the prevalence of glomerular capillary thrombi.	[83]
1998	Anti-GBM nephritis mice model	*Tnf*-α knockout mice	In TNF-deficient mice, the influx of lymphocytes was reduced, the development of proteinuria was delayed and the formation of crescents was almost completely prevented.	[87]
1999	UUO mice model	*Tnfr*1 and *Tnfr*2 knockout	Individual knockout of the TNFRs genes resulted in significantly less NF-kB activation compared with the WT. *Tnfr*1 knockout showed a significant reduction in *Tnf-α* mRNA levels compared with WT or *Tnfr*2 knockout mice.	[93]
2001	Rat model of crescentic glomerulonephritis	TNF-α blockade with sTNFR1	Treatment with sTNFR1 caused a marked reduction in albuminuria, reduced glomerular cell infiltration, activation, and proliferation, and prevented the development of crescents.	[88]
2003	Mice model of cisplatin-induced acute renal failure	*Tnfr*1 and *Tnfr*2 knockout	*Tnfr*2-deficient mice developed less-severe renal dysfunction and showed reduced necrosis, apoptosis, and leukocyte infiltration into the kidney, and lower renal and serum TNF levels compared with either *Tnfr*1-deficient or WT mice.	[94]
2003	Streptozotocin (STZ)-induced diabetic rats	Administration of a TNF antagonist (TNFR:Fc)	Administration of a TNF antagonist reduces urinary TNF-α excretion and prevents sodium retention and renal hypertrophy. TNF-α contributes to early diabetic nephropathy, and its inhibition may attenuate early pathological changes.	[98]
2005	Rat model of nephrotoxic nephritis	Administration of anti-TNF-α antibody	Neutralization of endogenous TNF-α reduces glomerular inflammation, crescent formation, and tubulointerstitial scarring, with preservation of renal function.	[103]
2005	Anti-GBM nephritis mice model	*Tnfr*1 or *Tnfr*2 knockout	Lack of Tnfr1 resulted in excessive renal T cell accumulation and an associated reduction in apoptosis of these cells. *Tnfr*2-deficient mice were completely protected from glomerulonephritis, despite an intact systemic immune response.	[92]
2005	UUO rat model	TNF-α blockade with PEG-sTNFR1	Treatment with PEG-sTNFR1 reduced tissue Tnf-α and protein production, renal tubular cell apoptosis, and caspase activity.	[89]
2007	UUO rat model	TNF-α blockade with PEG-sTNFR1	Renal obstruction induced increased tissue TNF-α and several markers of renal fibrosis, whereas treatment with PEG-sTNFR1 significantly reduced each of these markers of renal fibrosis.	[90]
2007	Rat model of kidney transplantation	Treatment with cyclsporine	In rats with acute allograft rejection, significantly elevated expression of TNFR2 was observed in tubular epithelial cells, podocytes, B cells, and monocytes/macrophages. TNFR2 expression levels were associated with renal function.	[104]
2007	STZ-induced diabetic rats	Administration TNF-α inhibitors, Infliximab and FR167653	TNF-α inhibition with infliximab and FR167653 decreased urinary albumin excretion, suggesting the role of TNF-α in the pathogenesis of diabetic nephropathy, with TNF-α inhibition is a potential therapeutic strategy.	[100]
2008	UUO mice model	*Tnf-α* knockout	*Tnf*-deficient mice showed an increase of extracellular matrix in the kidneys and infiltrating macrophages, explained by the increased TNFR2 expression level.	[105]
2009	SLE prone mice models	*Tnfr*1*, Tnfr*2 and double *Tnfr*1*/*2 knockout	Doubly-deficient mice developed accelerated pathological and clinical nephritis, while mice deficient in either TNFR, alone, did not differ from each other or from WT controls.	[106]
2010	ANG II-dependent mice model of hypertension	*Tnfr*1 knockout	Angiotensin II inhibited renal *Tnfr*1 mRNA accumulation, while increasing that of *Tnfr*2. Deletion of *Tnfr*1 was associated with increased albuminuria and creatinine clearance, in response to ANG II infusion.	[91]
2013	Anti-GBM nephritis mice model	*Tnf-α, Tnfr*1 and *Tnfr*2 knockout	*Tnfr*2 deficiency resulted in a reduction in renal macrophage but not neutrophil accumulation, while *Tnfr*1 deletion prevented the influx of both leukocyte subsets.	[79]
2013	TNF-induced inflammation mice model	*Tnfr*1*, Tnfr*2 and double *Tnfr*1*/*2 knockout	TNF-induced glomerular leukocyte infiltration was abrogated in *Tnfr*1-deficient mice, whereas *Tnfr*2-deficiency decreased mononuclear phagocytes infiltrates, but not neutrophils.	[107]
2014	Mice models of LPS- or TNF-induced acute endotoxemia	*Tnfr*1 knockout	LPS and TNF-treated WT models showed alterations of glomerular endothelium, increased albuminuria, and decreased GFR. The effects of LPS on the glomerular endothelial surface layer, GFR, and albuminuria were diminished in *Tnfr*1 knockout mice.	[108]
2014	Type 2 diabetic model of the KK-Ay mouse	TNF-α inhibition with Etanercept (ETN)	Renal mRNA and/or protein levels of *Tnfr*2, but not *Tnf-α* and *Tnfr*1, in ETN-treated mice were significantly decreased. ETN may exert a renal protective effect via inhibition of the inflammatory pathway activated by TNFR2 rather than TNFR1.	[99]
2017	Mice with CaOx nephrocalcinosis-related CKD	*Tnfr*1*, Tnfr*2 and double *Tnfr*1*/*2 knockout	WT mice developed progressive CKD, while *Tnfr*1*-*, *Tnfr*2*-*, and *Tnfr*1*/*2-deficient mice lacked intrarenal CaOx deposition and tubular damage, despite exhibiting similar levels of hyperoxaluria.	[95]
2019	STZ-induced diabetic rats	Treatment with adalimumabe, a TNF-α inhibitor	TNF-α inhibition reduced albuminuria, glomerular injury, and tubular injury in STZ-induced diabetic rats. TNF-α inhibition reduced the NLRP3 inflammasome in tubules and decreased expression of tubular IL-6 and IL-17A mRNA.	[97]
2020	Rodent models of 2,8-DHA crystal nephropathy	*Tnfr*1 and *Tnfr*2 knockout	Deletion of *Tnfr*1 significantly reduced tubular inflammation, thereby ameliorating the disease course. In contrast, genetic deletion of *Tnfr*2 had no effect on the manifestations of 2,8-DHA nephropathy.	[96]
2021	Ischemia-reperfusion mice model	Clamping of the renal pedicles	Proximal tubular cells exhibited a profibrotic and proinflammatory profile, and a marked transcriptional activation of NF-κB and AP-1 signaling pathways.	[102]

Abbreviations: ANG II, angiotensin II; AP-1, activator protein 1; CaOx, calcium oxalate; DHA, 2,8-dihydroxyadenine; ETN, Etanercept; GBM, glomerular basement membrane; LPS, lipopolysaccharide; NF-κB: factor nuclear kappa B; SLE, systemic lupus erythematosus; sTNFR1, soluble tumor necrosis factor receptor 1; STZ, Streptozotocin; TNFR1, tumor necrosis factor receptor 1; TNFR2, tumor necrosis factor receptor 2; TNF-α, tumor necrosis factor alpha; UUO, unilateral ureteral obstruction; WT, wild type.

### 4.2. Studies Addressing TNF-α and TNFRs in Human Kidney Disease and Related Clinical Outcomes

In clinical studies, circulating TNFR1 and TNFR2 were shown to be increased in several cohorts of patients, with different CKD etiologies and diverse age-groups and races (Table 2). Despite being responsible for engaging different downstream signaling pathways, the strength of associations with renal function is similar for both receptors.

The first study assessing the serum levels of a TNFR (unidentified either as TNFR1 or TNFR2) in CKD patients was published in 1994, and showed a strong correlation between the receptor levels and serum creatinine, in a group of 26 non-dialyzed CKD patients [109]. TNFR1 and TNFR2 were further associated with eGFR and with albuminuria in several subsequent studies [110,111,112,113]. In a prospective cohort that included 984 CKD patients, eGFR was negatively correlated with the serum levels of TNFR1 and TNFR2 [113]. To a lesser extent, both biomarkers were also positively correlated with urinary protein-to-creatinine ratio. Furthermore, in a cohort of patients, with a diverse set of kidney diseases and undergoing native kidney biopsy, TNFR1 and TNFR2 plasma levels were associated with underlying histopathologic lesions and adverse clinical outcomes, such as disease progression and death [114].

In a group of 106 biopsy-proven IgA nephropathy patients, higher serum levels of TNFR1 and TNFR2 were present in patients with more severe renal interstitial fibrosis [112]. Increased circulating levels of TNF receptors were similarly described as prognostic markers of idiopathic membranous nephropathy [115] and contrast-induced nephropathy [116]. Patients with systemic lupus erythematosus (SLE) have also been studied, with urinary TNFR1 [117] and serum TNFR2 [118] levels being elevated in cases of lupus nephritis.

The predictive value of TNFRs was mostly described in diabetic nephropathy, as reviewed by Murakoshi et al. (2020) [119]. Several results from the Joslin Kidney Center studies showed that the TNFRs seem to be candidate biomarkers of renal function decline in both type 1 [120] and type 2 [121] diabetic patients. Moreover, in type 1 diabetic patients, the increased circulating levels of TNFR1 and TNFR2 were the strongest determinants of CKD progression, preceding the onset of microalbuminuria and/or its progression to macroalbuminuria [122]. Higher baseline circulating levels of TNFR1 and TNFR2 were associated with a higher risk of eGFR worsening in patients with both early and established diabetic nephropathy [123]. A systematic review and meta-analysis highlighted the reliability of TNFRs in predicting diabetic kidney disease progression. The results seem to be consistent across different cohorts of diabetic patients [124,125,126,127]. A recently published study, evaluated a composite risk score termed KidneyIntelX for predicting the progression of diabetic kidney disease, in a large multinational cohort. KidneyIntelX comprises clinical variables and the circulating levels of three biomarkers, TNFR1, TNFR2, and kidney injury molecule 1 (KIM-1). KidneyIntelX successfully stratified patients for disease progression, showing that, after 1 year, a greater reduction in eGFR was observed in patients with higher changes in KidneyIntelX risk scores, independently of the baseline risk score value and the treatment option [128].

CKD patients have an increased risk of mortality due to CVD, which is independent of the traditional risk factors, possibly due to the chronic inflammatory state. Both circulating TNFRs were described as predictors of CVD risk [113,129] and all-cause mortality [129,130] in CKD populations, independently of eGFR and albuminuria, and irrespective of the cause of kidney disease. Some studies [131,132,133] have also addressed the prognostic value of circulating TNFRs in HD patients. Despite TFNRs being substantially linked with other inflammatory markers, Carlsson et al. observed no significant connection between either TNFRs and death, in a longitudinal cohort analysis of 207 prevalent HD patients [131]; two more recent studies, including one from our team, reported that circulating levels of TNFR1 and TNFR2 are independent predictors of all-cause mortality in ESKD patients under chronic HD [132,133] (REF 2017 and 2021), although for cardiovascular mortality, the significance was only observed for TNFR1 [132].

In the last 15 years, proteomic and transcriptomic studies have proven useful in discovering new insights into the TNF-α signaling pathway in CKD, as well as the associated-comorbidities. In a proteomic analysis of human serum from patients with CKD, TNF-α was associated with disease severity [134,135], as well as with vascular changes [134]. The circulating extracellular vesicles of CKD patients showed a pro-inflammatory profile, that included markers of the TNF signaling pathways. Niewczas et al. measured 194 circulating inflammatory proteins using aptamer-based proteomics analysis of different cohorts of diabetic patients [136]. The results showed that, out of the 194 measured proteins, 17 were TNFR superfamily-related, and also that TNFR1 and TNFR2 were strong predictors of renal function decline [136]. Accordingly, Ihara et al. showed that a profile of multiple circulating TNF receptors, including TNFR1 and TNFR2, was associated with early progressive renal decline in type 1 diabetes [137]. Tubular cells of IgA nephropathy patients also overexpressed genes of the inflammatory TNF signaling pathway [138].

### 4.3. Anti-TNF-α Theraphy in Patients with Impaired Kidney Function

The huge amount of scientific evidence linking TNF signaling with the pathophysiology of CKD raises questions regarding the utility and safety of therapeutic strategies targeting TNF-α in humans with impaired kidney function.

Nephrotoxicity is a rare side effect of anti-TNF-α medications, and a few reports of this occurrence have been described in the literature. Premužić et al. reported an association of TNF-α inhibitors (adalimumab and golimumab) and the development of IgA nephropathy in three patients with both rheumatoid arthritis and diabetes, but without history of renal disease [139]. Moreover, Stokes et al. showed that a subset of patients on anti-TNF-α therapy, who had no prior evidence of renal diseases, developed glomerulonephritis. This was supported by serologic abnormalities and by the presence and formation of auto-antibodies [140].

However, other studies demonstrated the therapeutic benefit of TNF-α blocking in improving renal inflammation and function. In patients with rheumatoid arthritis and CKD, the administration of anti-TNF-α was associated with less renal function decline [141]. In addition, the use of anti-TNF-α agents showed promising results in renal vasculitis [142] and kidney transplant recipients with rheumatic disease [143].

There is a limitation to the beneficial effects of anti-TNF-α agents, which seems to be related to their ability to induce autoimmunity by disrupting TNF-α normal immune regulation. Their use in clinical practice would require surveillance for complications. Indeed, the biological functions of cytokines are complex, and, thereby, blocking of cytokines might induce other unexpected and unclear effects in vivo. Furthermore, the effects of anti-TNF-α agents might be modulated by other factors, such as their distribution into diseased tissues, and degradation by proteases. Given the potential benefits of these therapies, a deeper understanding of the TNF signaling pathway and the mechanisms of action of the anti-TNF-α agents and their correlation with the clinical settings is needed for a more appropriate and personalized selection of therapeutic agents, and even for the development of new biological preparations, to be applied in the treatment of inflammatory diseases.

**Table 2 ijms-23-03284-t002:** Association of TNF-α and TNF receptors with renal dysfunction and disease, as well as with adverse clinical outcomes in human studies.

Year	Study Type	Study Population	Biomarkers	Study Outcomes	Reference
1994	Cross-sectional	26 non-HD CKD patients, 61 HD patients, 43 renal transplant recipients and 34 healthy controls	Serum levels of TNFR?	All patient groups showed significantly higher TNFR levels compared to the control group. A correlation of TNFR and creatinine levels was only found in the group of non-dialyzed CKD patients.	[109]
2005	Retrospective cohort	687 individuals from the CARE trial study, with CKD and previous myocardial infarction	Serum levels of TNFR2	Higher TNFR2 is independently associated with faster rates of kidney function loss in CKD. Inflammation may mediate the loss of kidney function among subjects with CKD and concomitant coronary disease.	[110]
2007	Cross-sectional	38 patients with SLE and 15 healthy controls	Urinary levels of TNFR1	Urinary TNFR1 levels were elevated in patients with lupus nephritis and correlated with proteinuria and SLE disease activity index scores.	[117]
2007	Prospective cohort	3075 adults aged 70 to 79	Serum levels of TNF-α, TNFR1 and TNFR2	In an elderly cohort of patients with eGFR ≥ 60 mL/min/1.73 m^2^, cystatin C was strongly associated with TNF-α and the TNFRs.	[144]
2008	Cross-sectional	6814 participants free of cardiovascular disease, from the MESA study	Circulating levels of TNFR1	Creatinine-based eGFR had significant correlations with TNFR1, in both participants with and without CKD.	[145]
2009	Cross-sectional	96 human renal allograft biopsies	Renal TNFR2 expression	In human renal transplant biopsies, there was an increase in the number of TNFR2-positive podocytes, in tubular epithelial cells, B cells, and monocytes/macrophages.	[104]
2009	Cross-sectional	667 participants with diabetes	Serum levels of TNF-α, TNFR1 and TNFR2	Elevated concentrations of serum markers of the TNF-α pathway were strongly associated with decreased renal function in T1D patients without proteinuria.	[146]
2010	Prospective cohort	55 patients with biopsy-proven primary glomerulonephritis and 20 healthy controls	Urinary levels of TNFR1	Elevated TNFR1 urinary levels predicted renal function decline and advanced renal interstitial fibrosis in patients with primary nephropathy.	[147]
2010	Cross-sectional	3294 participants from the Framingham Offspring Study, 291 of them with CKD	Serum levels of TNF-α and TNFR2	A significant proportion of variability in TNFR2 concentration was explained by CKD status and higher cystatin C quartiles. Higher concentrations of TNF and TNFR2 were associated with CKD status, higher cystatin C, and higher UACR.	[148]
2011	Prospective cohort	4926 patients followed for 15 years	Serum levels of TNFR2	For the risk of developing incident CKD among those who were CKD-free at baseline, only TNFR2 and IL-6 levels, but not CRP, were positively associated with incident CKD.	[149]
2012	Prospective cohort	3939 participants with established CKD	Plasma levels of TNF-α	Biomarkers of inflammation (cytokines and acute phase proteins) were higher in participants with lower levels of kidney function and higher levels of albuminuria.	[9]
2012	Prospective cohort	628 patients with T1D, normal renal function, and no proteint2uria	Serum levels of TNFR1 and TNFR2	Elevated serum concentrations of TNFR1 and TNFR2 were strongly associated with early renal function loss, progression to CKD stage 3 or higher, in patients with T1D who had normal renal function.	[120]
2012	Prospective cohort	410 patients with T2D	Serum levels of TNFR1 and TNFR2	Elevated concentrations of circulating TNFRs in patients withT2D at baseline were very strong predictors of the subsequent progression to ESRD in subjects with and without proteinuria.	[121]
2012	Prospective cohort	12 patients with active lupus nephritis, 14 with inactive SLE, and 14 healthy subjects	Serum levels of TNF-α and TNFR2	TNFR2 serum levels were elevated in all patients with active lupus nephritis and declined after clinical remission.	[118]
2013	Prospective cohort	84 glomerulonephritis patients under immunosuppressive therapy and 18 healthy controls	Serum and urine levels of TNFR1 and TNFR2	Urinary levels, but not serum levels, of TNFR1 and TNFR2 were effective in predicting a favorable response to immunosuppressive treatment in patients with primary glomerulonephritis.	[150]
2014	Prospective cohort	Patients with T1D and normoalbuminuria (286) or microalbuminuria (248)	Serum levels of TNF-α, TNFR1 and TNFR2	In both groups, the strongest determinants of renal decline were baseline serum concentrations of uric acid and TNFRs. Renal decline was not associated with sex or baseline serum concentration of the other measured markers.	[122]
2014	Prospective cohort	113 patients with biopsy-proven iMN and 43 healthy volunteers	Serum levels of TNFR1 and TNFR2	Estimated glomerular filtration rate and proteinuria tended to worsen as the TNFRs levels increased. Renal tubular TNFRs expression was associated with circulating TNFRs levels.	[115]
2014	Prospective cohort	522 T2D patients with DKD	Serum levels of TNFR1	TNFR1 is a strong prognostic factor for all-cause mortality in T2D with renal dysfunction, and its clinical utility is suggested in addition to established risk factors for all-cause mortality.	[130]
2014	Prospective cohort	429 patients with T1D and overt nephropathy	Plasma levels of TNFR1	Circulating levels of TNFR1 were highly correlated with eGFR, especially in patients with an eGFR < 60 mL/min/1.73 m^2^. Circulating levels of the TNFR1 also remained associated with ESRD after adjusting for the competing risk of death.	[151]
2014	Prospective cohort	349 T1D patients with proteinuria and CKD staged 1–3	Serum levels of TNFR2	Serum TNFR2 was the strongest determinant of renal decline and ESRD risk. The rate of eGFR loss became steeper with rising concentration of TNFR2, and elevated HbA1c augmented the strength of this association.	[152]
2015	Prospective cohort	223 biopsy-proven primary IgA nephropathy patients	Serum levels of TNFR1 and TNFR2	Both TNFRs levels were significantly higher in patients with eGFR < 60 mL/min/1.73 m^2^ than in patients with higher eGFR. Both TNFRs were associated with renal function decline, independent of age and uric acid levels.	[153]
2015	Prospective cohort	262 patients admitted for a CAG and/or a PCI	Serum levels of TNFR1 and TNFR2	Markedly elevated concentrations of circulating TNFRs were correlated with the occurrence of contrast-induced nephropathy (CIN) and significantly associated with prolonged renal dysfunction, regardless of the development of CIN.	[116]
2015	Prospective cohort	131 patients with CKD at stages 4 and 5	Serum levels of TNFR1 and TNFR2	Both TNFRs were independently associated with all-cause mortality or an increased risk for cardiovascular events in advanced CKD, irrespective of the cause of kidney disease.	[129]
2015	Prospective cohort	347 patients with newly diagnosed biopsy-proven primary IgA nephropt2athy	Plasma levels of TNFR1 and TNFR2	eGFR decreased and proteinuria worsened proportionally as TNFR1 and TNFR2 levels increased. Tubulointerstitial lesions, such as interstitial fibrosis and tubular atrophy, were significantly more severe as concentrations of circulating TNFRs increased, regardless of eGFR levels.	[154]
2015	Prospective cohort	193 Pima Indians with T2D	Serum levels of TNFR1 and TNFR2	Elevated serum concentrations of TNFR1 or TNFR2 were associated with increased risk of ESRD in American Indians with type 2 diabetes, after accounting for traditional risk factors including UACR and mGFR.	[155]
2015	Cross- sectional	106 biopsy-proven IgA nephropathy patients and 34 healthy subjects	Serum and urinary levels of TNFR1 and TNFR2	Elevated serum TNFR1 or TNFR2 levels were significantly associated with the severity of renal interstitial fibrosis after adjusting for eGFR, UPCR, and other markers of tubular damage.	[112]
2015	Prospective cohort	207 patients undergoing HD	Serum levels of TNFR1 and TNFR2	Prevalent hemodialysis patients had several-fold higher levels of sTNFRs compared to previous studies in CKD stage-4 patients. However, no consistent association between TNFR and mortality was observed.	[131]
2016	Prospective cohort	2220 Chinese patients aged 50–70 years old with eGFR > 60 mL/min/1.73 m^2^	Plasma levels of TNFR2	Elevated levels of TNFR2 were independently associated with a greater risk of kidney function decline in middle-aged and elderly Chinese.	[156]
2016	Prospective cohort	83 Pima Indians with T2D	Serum levels of TNF-α, TNFR1 and TNFR2	TNFR1 and TNFR2 significantly correlated inversely with the percentage of endothelial cell fenestration and the total filtration surface per glomerulus. Thus, TNFRs may be involved in the pathogenesis of early glomerular lesions in DN.	[124]
2016	Prospective cohort	86 patients with CKD stages 2–4	A panel of biomarkers, including TNF-α	The panel of proteomic inflammatory and mineral and bone disorder biomarkers showed a better performance in detecting early CKD stages, disease progression, and vascular changes, than each single biomarker.	[134]
2016	Prospective cohort	3430 participants with eGFR of 20–70 mL/min/1.73 m^2^	Plasma levels of TNF-α	Elevated plasma levels of TNF-α and decreased serum albumin were associated with rapid loss of kidney function in patients with CKD.	[10]
2016	Prospective cohort	543 patients with stage 5 CKD	Serum levels TNF-α	TNF-α could, independently of other biomarkers, predict all-cause mortality, but not clinical CVD.	[157]
2016	Prospective cohort	607 Swedish patients with T2D	Circulating levels of TNFR1 and TNFR2	Higher levels of both TNFR1 and TNFR2 were associated with prevalent diabetic kidney disease, as well as with worsened kidney function and higher urinary albumin/creatinine ratio.	[127]
2017	Nested case-control	380 participants with early DKD (190 matched case-control pairs) from the ACCORD study	Plasma levels of TNFR1 and TNFR2	At baseline, median levels of TNFR1 and TNFR2 were roughly two-fold higher in the advanced than in the early cohort. TNFR1 and TNFR2 levels were associated with higher risk of eGFR decline in T2DM persons with both early (ACCORD) and established (VA-NEPHRON-D) DKD. In both cohorts, patients who reached the renal outcome had higher baseline TNFRs levels.	[123]
	Prospective cohort	1256 participants with advanced DKD from the VA-NEPHRON-D Cohort			
2017	Prospective cohort	984 patients with CKD	Serum levels of TNFR1 and TNFR2	TNFR1 and TNFR2 predicted CVD risk, even after adjustment for clinical covariates, such as urinary protein/creatinine ratio, eGFR, and high-sensitivity CRP.	[113]
2017	Prospective cohort	1.135 French patients with T2D	Serum levels of TNFR1	In addition to established risk factors, TNFR1 improves risk prediction of loss of renal function in patients with T2D.	[125]
2017	Prospective cohort	319 patients receiving maintenance hemodialysis	Serum levels of TNF-α, TNFR1 and TNFR2	Elevated TNFRs levels were associated with an increased risk of cardiovascular and/or all-cause mortality, independently of other studied covariates, in patients undergoing HD.	[132]
2017	Prospective cohort	122 patients with confirmed DN	Renal tissue expression of TNFR1 and TNFR2	No correlations were found between glomerular or tubular expressions of TNFRs, and clinical parameters, including GFR decline slopes.	[158]
2018	Prospective cohort	453 Indigenous Australians with and without diabetes and/or CKD	Serum levels of TNFR1	Circulating levels of TNFR1 were associated with greater kidney disease progression, independently of albuminuria and eGFR, in Indigenous Australians with diabetes.	[126]
2018	Prospective cohort	594 Japanese patients with T2D and eGFR > 30 mL/min/1.73 m2 (stages 1 to 3)	Serum levels of TNF-α, TNFR1 and TNFR2	Circulating TNF-related inflammatory biomarkers were associated with urinary albumin/creatinine ratio and eGFR. Among the biomarkers, the association of TNFRs with eGFR was the strongest after adjustment for relevant covariates.	[111]
2018	Prospective cohort	2399 patients with CKD and no history of cardiovascular disease	Plasma levels of TNF-α	A composite inflammation score with 4 biomarkers (IL-6, TNF-a, fibrinogen, and albumin) was associated with a graded increase in risk for incident atherosclerotic vascular disease events and death in patients with CKD.	[23]
2018	Prospective cohort	2871 participants multiethnic cohort	Serum levels of TNFR1	Elevated serum TNFR1 concentrations were associated with faster declines in eGFR over the course of a decade in a multiethnic population, independently of previously known risk factors for kidney disease progression.	[159]
2019	Prospective cohort	525 diabetic participants of 3 independent cohorts	194 proteins, including TNFR1 and TNFR2	Kidney risk inflammatory signature (KRIS) comprising 17 circulating inflammatory proteins, including TNFR1 and TNFR2, were associated with incident ESRD in diabetic patients.	[136]
2019	Systematic review and Meta-analysis	6526 participants from 11 cohorts for TNFR1 measurements and 5385 participants from 10 prospective for TNFR2 measurements	Circulating levels of TNFR1 and TNFR2	Circulating TNFR-1 and TNFR-2 are reliable predictors of DKD progression.	[160]
2019	Prospective cohort	47 patients with diabetes and eGFR > 60 mL/min/1.73	Serum levels of TNFR1	In patients with an early decline in renal function, TNFR1 values increased as eGFR decreased, over an 8-year period. In contrast, there were no significant changes in soluble TNFR1 levels in patients with stable renal function.	[161]
2020	Prospective cohort	165 case participants from the ADVANCE trial and 330 matched control	Plasma levels of TNFR1 and TNFR2	Elevated circulating TNFR1 and TNFR2 levels were associated with poor kidney outcome.	[162]
2020	Cross-sectional	26 adults with terminal stage CKD and 10 healthy controls	Serum levels of 27 cytokines, including TNF-α	Serum levels of TNF-α were increased 6 to 12 times in patients with CKD, as compared to controls. TNF-α levels positively correlated with complement systems components.	[135]
2020	Prospective cohort	894 CRIC Study participants with diabetes and an eGFR of < 60 mL/min/1.73 m^2^	Plasma levels of TNFR1 and TNFR2	Higher plasma levels of TNFR1 and TNFR2 were associated with increased risk of progression of DN. TNFR2 had the highest risk after accounting for the other biomarkers.	[163]
2020	Prospective cohort	651 children with 1–16 years old with an eGFR of 30–90 mL/min/1.73 m^2^	Plasma levels of TNFR1 and TNFR2	Children with a plasma TNFR1 or TNFR2 concentration in the highest quartile were at significantly higher risk of CKD progression, compared with children with a concentration for the respective biomarker in the lowest quartile.	[164]
2021	Prospective cohort	139 adults with CKD stages 1 to 5	Serum levels of 11 markers, including TNFR1 and TNFR2	Patients with high TNFR1, coupled with low complement 3a desarginine, almost universally (96%) developed the composite renal and mortality endpoint.	[165]
2021	Prospective cohort	346 T1D patients, 198 with macroalbuminuria and 148 with microalbuminuria	25 TNF family proteins, including TNFR1 and TNFR2	Levels of TNR1 and TNFR2 were associated with increased risk of early progressive renal decline in T1D diabetic patients with macro and microalbuminuria.	[137]
2021	Prospective cohort	523 CKD patients undergoing kidney biopsy with a diverse set of kidney diseases	Plasma levels of TNFR1 and TNFR2	Both TNFR1 and TNFR2 were associated with tubulointerstitial and glomerular lesions; each doubling of TNFR1 and TNFR2 was associated with an increased risk of CKD progression, but only TNFR2 was associated with risk of death.	[114]
2021	Prospective cohort	2553 patients with T2D and normoalbuminuria	Plasma levels of TNFR1 and TNFR2	Each doubling of baseline TNFR1 and TNFR2 was associated with a higher risk of kidney outcome (40% reduction in eGFR or kidney failure), in normoalbuminuric patients.	[166]
2021	Prospective cohort	3523 participants from the CANVAS placebo-controlled trial	Plasma levels of TNFR1 and TNFR2	Each doubling in baseline TNFR1 and TNFR2 was associated with a higher risk of kidney outcomes. Early decreases in TNFR1 and TNFR2 during treatment were associated with a lower risk of disease progression.	[167]
2021	Cross-sectional	499 patients with T2D and eGFR ≥ 60 mL/min/1.73 m^2^	Serum and urinary TNFR1 and TNFR2 levels	Kidney measures appear to be strongly associated with serum TNFRs, rather than urinary TNFRs in patients with type 2 diabetes and normal renal function.	[168]
2021	Prospective cohort	594 participants with T2D and eGFR < 60 mL/min/1.73 m^2^	Plasma levels of TNFR1 and TNFR2	TNFR1 and TNFR2 were associated with risk of incident kidney failure needing RRT, in adults with diabetes and an eGFR < 60 mL/min/1.73 m^2^, after adjustment for established risk factors.	[169]
2021	Cross-sectional	5 human renal biopsy specimens from IgA nephropathy patients and 1 healthy control	Transcriptomic analysis of single-cell RNA	Tubular cells of IgA nephropathy patients were enriched in inflammatory pathways, including TNF-α signaling.	[138]
2021	Prospective cohort	289 ESRD patients under chronic HD therapy	Several biomarkers circulating levels, including TNFR2	TNFR2 levels were an independent predictor of all-cause mortality (1-year follow-up study). Circulating levels of cfDNA emerged as the best predictor of mortality.	[133]
2022	Prospective cohort	1325 participants from the CANVAS trial with prevalent DKD	KidneyIntelX score, including plasma levels of TNFR1 and TNFR2	Changes in the KidneyIntelX score from baseline to 1 year were associated with future risk of CKD progression, independently of the baseline risk score and treatment arm.	[128]

Abbreviations: ACCORD, Action to Control Cardiovascular Risk in Diabetes trial; ADVANCE, Action in Diabetes and Vascular Disease; CAG, coronary angiography; CARE, The Cholesterol and Recurrent Events trial; cfDNA, cell-free DNA; CIN, contrast-induced nephropathy; CKD, chronic kidney disease; CRIC, chronic Renal Insufficiency Cohort; CRP, C-reactive protein; CVD, cardiovascular disease; DN, diabetic nephropathy; eGFR, estimated glomerular filtration rate; ESRD, end-stage renal disease; Hba1c, hemoglobina glicada; HD, hemodialysis; IgAN, IgA nephropathy; IL-6, Interleukin-6; iMN, idiopathic membranous nephropathy; KRIS, kidney risk inflammatory signature; MESA, Multi-ethnic study of atherosclerosis; mGFR, measured glomerular filtration rate; PCI, percutaneous coronary intervention; SLE, systemic lupus erythematosus; T1D, type 1 diabetes; T2D, type 2 diabetes; TNFR1, tumor necrosis factor receptor 1; TNFR2, tumor necrosis factor receptor 2; TNF-α, tumor necrosis factor alpha; UACR, urinary albumin-to-creatinine ratio; UPCR, urinary protein-to-creatinine ratio; VA-NEPHRON, Veterans Administration NEPHROpathy iN Diabetes study.

## 5. Considerations for Future Research

Despite recent breakthroughs in CKD care, the rates of morbidity and mortality are still unacceptable. Chronic inflammation is a common feature in kidney diseases, regardless of its etiology, and which plays a key role in disease pathophysiology, progression, and development of associated complications. Unresolved inflammatory processes generally lead to renal fibrosis and ESRD.

The role of TNF-α in the pathogenesis of kidney diseases depends on the engagement of receptor-specific and/or common signaling cascades. The differential expression of both receptors in different cell types, and the fact that soluble and transmembrane TNF-α present different affinities to each receptor, are other factors that contribute to the complexity of TNF-α signaling.

Circulating TNFRs have been associated with renal damage in several animal and human studies. Based on the available data, increased levels of TNFRs associate with decreased eGFR and increased albuminuria. Overall, TNFRs have proven to be useful and effective in predicting renal function decline and CKD progression, as well as CKD-associated morbidity and mortality, among different cohorts of patients in both cross-sectional and longitudinal studies. The consistency of the published literature evidences their potential role as prognostic and risk-predictive biomarkers in CKD, along with the traditional markers already used in clinical practice.

The mechanisms by which the TNFRs initiate and perpetuate renal damage are not completely understood. In fact, there is evidence that TNF-α is not the only molecule involved in the regulation of its receptors during renal function decline [84], suggesting that other molecules and chemokines act as potential downstream effectors on TNFRs. Moreover, the interplay between TNFR1 and TNFR2, the role of each receptor in specific kidney diseases (particularly in more rare diseases), and their prognostic value in patient outcomes deserve further investigation.

To date, there are no anti-inflammatory treatments for CKD patients. Treating inflammation and preventing the progression of renal fibrosis is complex, due to the crosstalk between the inflammatory signaling pathways. The approved therapeutic use of anti-TNF monoclonal antibodies is currently limited to autoimmune diseases, such as rheumatoid arthritis, Chron’s disease, or psoriatic arthritis [170]. Considering the relevance of the TNF signaling pathways in CKD pathophysiology, studies on the efficacy of the existing TNF biologics in renal diseases would be useful. Furthermore, individual inhibition of TNFR1 or TNFR2 may further clarify the balance of proinflammatory/immunomodulatory roles for each of these receptors.

Future research should focus on validating the promising findings in large, multicentered studies, with standardized methodologies, to allow their translation into clinical practice. TNFRs could be important tools to improve CKD patient’s characterization and management, with direct implications for strategies to prevent or postpone the progression of CKD. This may possibly result in a better prognosis for patients, as well as in financial benefits; lowering healthcare costs in CKD management.

## Figures and Tables

**Figure 1 ijms-23-03284-f001:**
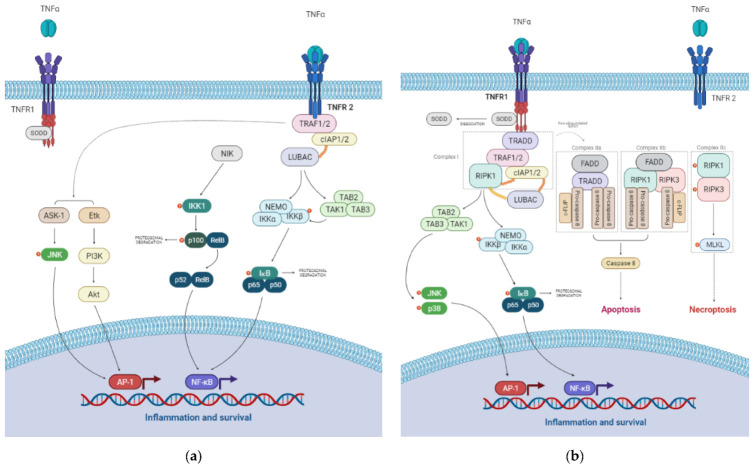
TNFR1 (**a**) and TNFR2 (**b**) mediated signaling pathways. Akt, protein kinase B; AP-1, activator protein-1; ASK-1, apoptosis signal-regulating kinase-1; c-FLIPʟ, cellular FLICE-inhibitory protein; cIAP1/2, cellular inhibitor of apoptosis protein 1 or 2; Etk, endothelial/epithelial protein tyrosine kinase; FADD, Fas-associated death domain; IKK, inhibitor of kappa B kinase; IκB, NF-ĸB inhibitor; JNK, c-jun kinase; LUBAC, linear ubiquitin chain assembly complex; MAPK, mitogen activated protein kinase; MLKL, mixed lineage kinase domain-like protein; NEMO, NF-κB essential modulator; NF-ĸB, nuclear factor kappa B; NIK, NF-κB inducing kinase; PI3K, phosphatidylinositol 3-kinase; RIPK1/3, receptor interacting serine/threonine-protein kinase 1 or 3; SODD, silencer of death domains; TAB, TAK-binding proteins; TAK1, transforming growth factor-beta-activated kinase 1; TNFR1, tumor necrosis factor receptor 1; TNFR2, tumor necrosis factor receptor 2; TNF-α, tumor necrosis factor alpha; TRADD, TNF receptor-associated death domain; TRAF1/2, TNF receptor-associated factor 1 or 2.

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
