# Peer review of "The Signaling Pathway of TNF Receptors: Linking Animal Models of Renal Disease to Human CKD"

_ijms, 2022, doi:10.3390/ijms23063284_

Round 1

Reviewer 1 Report

In this review article,  Lousa and colleagues, concisely and comprehensively reviewed the relationship of singling pathway of TNF receptors in both animal kidney diseases models and patients cohort studies of CKD. They overviewed the major TNF-alpha signaling pathway through distinct two receptors (TNFR1 and TNNFR2).

Several points should be corrected before publication.

  1. Section title of 3.2 TNFR1 signaling pathway” is probably about “ TNFR2”.
  2. In page 12 , section 4.2. Studies addressing TNFα and TNFRs in human kidney disease and realted clinical outcomes

Sentence after line 338,

The predictive value of TNFRs was mostly described in diabetic nephropathy, as re-338 viewed by Murakoshi et al. (2020) [109].  

Reference number is supposed be 115)

Murakoshi, M.; Gohda, T.; Suzuki, Y. Circulating Tumor Necrosis Factor Receptors: A Potential Biomarker for the Progression of Diabetic Kidney Disease. International journal of molecular sciences 2020, 21, 1957, doi:10.3390/ijms21061957.

Also several mistakes of references are found and please review and  correct the references.

Reviewer 2 Report

In this manuscript, authors comprehensively reviewed tumor necrosis factor (TNF) receptor signaling, significance of TNF receptor signaling in animal models of kidney disease, and the association between circulating TNF receptor and kidney diseases in clinical trials. The subject of study seems to be interesting. However, there are some concerns in this review. The reviewer’s comments are described as follows.

1. This review is described based on many historical papers and includes relatively small number of recent papers. Recent “omics” techniques have clarified pathophysiological roles of a number of signaling molecules in kidney diseases. Thus, authors should review not only studies of circulating TNFR but also transcriptomic and proteomic studies including TNFR signaling in the kidney.

2. Regarding Table 1 and 2, authors should explain the methods by which they selected these articles. Whether they comprehensively listed all articles based on some keywords or they selected some articles based on specific criteria remained unclear.

3. In several case reports and case series, anti-TNF-alpha agents such as infliximab and etanercept have been associated with acute kidney injury and nephrotic syndrome. Are therapeutic strategies targeting TNF-alpha really feasible in clinical practice?

4. In line 95, authors suggested that inflammation induces epithelial to mesenchymal transdifferentiation (EMT) in the kidney. However, recent evidence has downgraded the importance of EMT from tubular cells to myofibroblasts in renal fibrosis. Intrinsic fibroblast activation and EndoMT have increasingly emphasized.

5. In line 208, TNFR1 should be corrected to TNFR2.

6. Regarding Figure 1, authors should explain how TNFR1 and TNFR2 signaling can be switched each other in the same cell. Can TNFR1 and TNFR2 be expressed in the same cells or different cells?

Round 2

Reviewer 2 Report

Authors have successfully addressed the reviewer's concerns in the revised manuscript. There are no more comments.